# Effects of Partial Replacement of Soybean with Local Alternative Sources on Growth, Blood Parameters, Welfare, and Economic Indicators of Local and Commercial Broilers

**DOI:** 10.3390/ani14020314

**Published:** 2024-01-19

**Authors:** Muazzez Cömert Acar, Berna Türkekul, Özlem Karahan Uysal, Sezen Özkan, Servet Yalcin

**Affiliations:** 1Department of Animal Science, Faculty of Agriculture, Ege University, 35100 İzmir, Türkiye; muazzez.comert@ege.edu.tr (M.C.A.); sezen.ozkan@ege.edu.tr (S.Ö.); 2Department of Agricultural Economics, Faculty of Agriculture, Ege University, 35100 İzmir, Türkiye; berna.turkekul@ege.edu.tr (B.T.); ozlem.uysal@ege.edu.tr (Ö.K.U.)

**Keywords:** alternative feedstuffs, meat-type chicken strains, growth performance, feather cleanliness, economic profitability

## Abstract

**Simple Summary:**

While the demand for poultry meat is increasing, arable land for crop production is limited. Therefore, the use of locally produced alternative sources in chicken diets has become a necessity. The substitution of soybean with local by-products such as sunflower meal, brewers’ dried grain, and wheat middlings or a combination of local by-products with black soldier larvae meal was evaluated in broilers. The substitution of soybean in broiler diets did not affect growth performance. However, the diet with black soldier fly larvae meal increased the production costs because of its high price. Therefore, lowering black soldier fly larvae meal price is the key issue hindering its inclusion in broiler diets.

**Abstract:**

The effects of the partial replacement of soybean with alternative local agri-industry by-products and black soldier fly (BSF) larvae meal on broiler growth performance, blood biochemistry, welfare, and, subsequently, economic performance of these diets were evaluated. A total of 524 day-old chicks from a local and a commercial strain were fed one of the three diets from the day of hatch to the slaughter age. The diets were the following: a soybean-based control diet, a diet in which soybean was partially replaced (SPR) with agri-industrial by-products, or a diet with BSF larvae meal added to the SPR (SPR + BSF). There was no effect of the diets on the slaughter weight, total feed consumption, and feed conversion of the chickens. The SPR + BSF diet reduced the blood glucose, alanine aminotransferase, aspartate aminotransferase, gamma-glutamyl transferase, protein, triglycerides, and cholesterol levels in the local chickens and the gamma-glutamyl transferase, protein, and creatinine levels in the commercial broilers. The negative effect of the SPR diet on plumage cleanliness in the commercial broilers was alleviated by the SPR + BSF diet, whereas 100% of the local birds presented either slight or moderate soiling. The results showed that, due to the high cost of the BSF larvae meal, the SPR + BSF diet was not economically feasible. In a further study, the price trends of BSF larvae will be examined from the standpoint of economic profitability conditions.

## 1. Introduction

Despite the negative impact of rising global feed prices on profitability, the market for chicken meat is projected to have steady growth due to its comparative advantages over beef and sheep meat in terms of affordability, health benefits, and environmental sustainability [1]. Soybean serves as the main source of plant protein utilized in conventional chicken diets. The increasing demand for soybean meal and its environmental impact has led to increased efforts to investigate alternative feedstuffs in chicken diet formulations [2]. Therefore, emphasis has been given to the use of locally available agro-industrial by-products that are not intended for human consumption and can be used in chicken nutrition [3]. Several previous studies have demonstrated the potential of the inclusion of by-products such as sunflower meal, brewers’ dried grain, and wheat middlings in broiler diets [4,5,6]. Although their use is limited due to their comparatively higher fiber and lower lysine and methionine levels than soybean meal, this limitation can be overcome by adding enzymes to the diets. The addition of enzymes has been shown to increase the nutritional value and digestibility of non-starch polysaccharides and proteins in broilers [7,8,9].

Sunflower meal is a byproduct of the sunflower seed oil extraction process. Moghaddam et al. [10] reported that the inclusion of up to 140 g/kg of SFM in the diet improved body weight gain, feed intake, feed conversion, and the weight of the gastrointestinal tract and gizzard and reduced blood low-density lipoproteins in broilers. Salari et al. [11] found that adding up to 210 g/kg full-fat sunflower seed to the broiler diet improved body weight gain, feed intake, and feed conversion ratios, while neither low-density lipoprotein nor blood alkaline phosphatase, Ca, P, glucose, and protein concentrations were affected.

Brewers’ dried grain is the most common by-product of the brewery industry. The chemical composition of this grain is subject to variation as a result of the brewing process, although it is known to contain proteins, fibers, and lipids [12]. The addition of 20% brewers’ dried grain in local chickens’ diets, partially replacing maize and soybean meal, contributed to gizzard development [7] and increased carcass yields and profit margins without affecting growth performance [13]. In contrast to these findings, the inclusion of brewers’ dried grain at a 3 to 9% concentration was reported to be optimum, while an improvement in kidney and liver functions was reported by reducing the blood AST levels and creatinine levels [14]. It was also shown that wheat middlings, a by-product of the wheat milling process, could be incorporated into broiler diets up to a 30% concentration without any detrimental effects on body weight gain, slaughter weight, and feed conversion ratio [15].

Recently, there has been growing interest in using insects as an alternative protein source to partially replace soybeans in broiler diets [2]. The European Commission authorized the use of processed animal proteins obtained from insects in poultry diets (Commission Regulation 2021/1372). The nutritional content of black soldier fly (*Hermetia illucens* L.) (BSF) larvae makes them suitable for meeting the dietary requirements of animals [16]. BSF larvae can be considered an environmentally sustainable alternative feedstuff. This is primarily due to the relatively small land area required to produce one kg of protein, their use of organic waste, and their low levels of greenhouse gas emissions [17,18,19,20]. The inclusion of 100 g/kg of BSF larvae meal has been shown to provide a protein supply covering 33.01% of the total protein requirements and improve the carcass weight of broilers without affecting their blood parameters and health status [21,22]. De Souza Vilela et al. [23] showed that a concentration of up to 20% of full-fat BSF larvae in broiler diets increased body weight gain compared to the control diet. Murawska et al. [24] concluded that the replacement of soybean meal with full-fat BSF larvae meal should be lower than 50% as higher inclusion levels of full-fat BSF meal deteriorated growth performance. Gariglio et al. [25] reported that the inclusion of de-fatted BSF larvae meal up to a 9% concentration in Muscovy ducks’ diets did not result in reduced growth and improved some of the blood traits. Moreover, it has been also suggested that the inclusion of whole larvae has been an effective mean of environmental enrichment for broiler chickens, positively affecting the welfare of birds by increasing foraging activity [26], improving footpad health, and reducing fear [27].

There is an increasing interest in using slower-growing and local chickens. Anadolu-T is a registered genotype for broiler production and is within the scope of the selection and breeding program in Türkiye. The pure lines of Anadolu-T and their crosses had lower body weights and impaired feed conversion ratios but a higher livability compared to commercial strains [28]. These chickens could be an alternative for niche markets and small local producers. On the other hand, feed is the main cost, with 65–75% of the total cost of broiler production. Therefore, the cost of a diet is also important, and the use of alternative protein sources depends not only on their impact on performance but also on their price. Based on the above-mentioned background, we hypothesized that locally produced agri-industrial by-products with enzyme and BSF larvae supplementation could be used to partially replace soybean in broiler diets. Therefore, the effect of diets in which soybean was partly replaced with sunflower meal, brewers’ dried grain, and wheat middlings as well as a combination of BSF larvae meal with these local agri-industrial by-products on broiler performance, blood biochemistry, animal welfare, and economic efficiency of production was investigated. A local and commercial line was used to evaluate the effects of the diets on the strains.

## 2. Materials and Methods

The present study is a part of the ongoing PRIMA project “Alternative animal feeds in Mediterranean poultry breeds to obtain sustainable products” (SUSTAvianFEED) (Grant number 2015). All procedures were approved by the Farm Animal Experiments Local Ethics Committee of the Faculty of Agriculture at Ege University (Approval no.: 2022/002, 7 April 2022).

### 2.1. Experimental Design and Diets

A total of 504 day-old chicks (252 chicks/strain) from local (Anadolu-T pure dam line) and commercial (Cobb 500) strains were used. The chicks from both sexes were individually weighed, wing-banded, and placed in 36-floor pens with 14 chicks in each pen in an environmentally controlled experimental house. Each 1.4 m × 1.2 m floor pen was furnished with wood-shaving litter, one round feeder, and nipple drinkers. The ambient temperature was adjusted to 32 ± 1 °C on the first day and gradually decreased to reach 24 ± 1 °C on day 21. The lighting program was 23L:1D for the first three days and then gradually reduced to 16L:8D (about 20 lx) by day seven, and kept until the end of the experiment.

The chicks were fed one of three diets with six replications from the day of hatching to the slaughter age, which was 40 d for the commercial and 55 d for the local strains. The control diet was a typical soybean–corn diet. Soybean partial replacement (SPR) was formulated by partly replacing soybean with local agri-industrial by-products consisting of high-protein sunflower meal, brewers’ dried grain, and wheat middlings. Dried BSF (5%) larvae meal was included in the SPR to obtain a SPR + BSF diet. An average of 24.9 and 42.2% of soybean meal was replaced with alternative agri-industrial by-products in the SPR and SPR + BSF diets, respectively. All the diets were formulated to be isocaloric and isonitrogenous. The proximate composition of the experimental diets and sunflower meal, brewers’ dried grain, wheat middlings, and dried BSF larvae meal reared on vegetable and bakery by-products in Türkiye (Germina Tarım Teknolojileri Tic. Ltd., Şti., Ankara, Türkiye) are shown in Table 1 and Table 2, respectively.

All the chicks were vaccinated against Newcastle disease, Gumboro disease, and infectious bronchitis at the hatchery. On days 10 and 18, Newcastle and infectious bronchitis vaccine recalls were performed. Coccidiostats were not used during the experimental period.

### 2.2. Measurements

Individual body weights were measured on days 0, 10, and 25 and at the slaughter age. Feed consumption was measured on the same dates on a pen basis. Feed conversion was calculated considering the weight of the dead birds.

At day 40 for the commercial and day 55 for the local chickens, eighteen broilers (three birds/replicate, nine from each sex) from each treatment close to the average weight were randomly selected. After 8 h of feed withdrawal, blood samples were collected in a tube during the slaughtering process. The breast (including the bones) and the legs (including the hip and the bones) were separated from the carcass of each bird and weighed. The weights of the liver, the whole intestine, the spleen, and the bursa of Fabricius were measured. The relative weights were calculated by dividing the individual organ weights by the live body weight of the birds. The serum was separated after centrifugation at 4 °C and 2750 rpm for 10 min and stored at −80 °C until the analysis of blood metabolites. Blood protein, triglycerides, uric acid, alanine aminotransferase (ALT), aspartate aminotransferase (AST), gamma-glutamyl transferase (GGT), creatinine, cholesterol, corticosterone, and the blood minerals of Ca, P, and Mg concentrations were measured using commercial kits (Mindray, China) and an autoanalyzer (Mindray BS-240 Vet, Nanshan, Shenzhen 518057, China). The ELISA kits (Chicken corticosterone ELISA kit; Bioassay Technology Lab., Shanghai, China) were used to assess and quantify the corticosterone concentration of the blood samples following the manufacturer’s instructions. The absorbance was measured at 450 nm, and the concentration of corticosterone was determined relative to the absorbance of the calibration curve.

Tonic immobility testing was conducted at 28 d of age and at the slaughter age. Two birds (one male and one female) from each replicate pen were randomly chosen (twelve in total), placed on their backs on a table, and held for 15 s. Then, hands were slowly removed from the birds, and the timer was turned on to measure the latency to the right, with a maximum of 120 s [29]. If the birds righted themselves in less than 15 s, the same procedure was repeated for up to three times. The birds were given a score of zero if all three attempts for TI induction failed. Longer tonic immobility durations were associated with a higher fearfulness [29,30]. On days 40 and 55, for the commercial and local chickens, respectively, plumage cleanliness and footpad dermatitis (FPD) were scored on all the birds. A four-point scoring scale for plumage cleanliness (0: intact; 1: slight soiling; 2: moderate soiling; and 3: severe soiling) [31] and a three-point scale for footpad dermatitis (0: intact footpad; 1: moderate FPD; and 3: severe FPD) were used [32].

### 2.3. Chemical Analysis

The diets were analyzed for dry matter, crude protein, ether extract, crude fiber, total starch, total sugar, and total phosphorus and calcium [33]. The metabolizable energy content (ME) of the diet was calculated from the chemical composition using the equation ME kcal/kg = 38 × [(Crude Protein) + (2.25 × Ether Extract) + (1.1 × Starch) + (1.05 × Total Sugar)] + 53 [32]. A Fiber-Tec system was used to determine the neutral detergent fiber (NDF) value with alpha amylase treatment and the acid detergent fiber (ADF) value [34]. Moreover, BSF larvae meal was analyzed for acid detergent insoluble nitrogen (ADIN) by Kheldahl [35] and for ash-free ADF in the residues of acid detergent fibers. The acid detergent fiber-linked protein (ADIP) was calculated by multiplying ADIN with 6.25. The chitin level in the BAF larvae meal, which was 8.32%, was calculated according to the following equation: Chitin (%) = ash free ADF (%)—ADIP (%) [36].

The fatty acid composition was determined using gas-chromatography after the lipid was extracted according to Folch et al. [37]. The analyses were performed on an Agilent 6890 Network Gas Chromatograph (USA). The column was an Agilent HP-88 capillary (100 m × 0.25 mm ID × 0.2 μm). An amino acid analysis was carried out via high-performance liquid chromatography using HPLC (Agilent 1260 Infinity II, Santa Clara, CA 95051, USA).

### 2.4. Cost–Benefit Analysis

The economic implications of substituting soybean meal with local ingredients and BSF larvae were evaluated using key criteria such as gross profit margin (GPM), cost–benefit ratio (CBR), and return on investment (RoI) [38]. The costs of feed consumed were determined using the market prices of the ingredients for 2023 and considering the quantities of each item used in the diets. The GPM was calculated using the following formula: GPM = total selling price of carcass (SP) − cost of feed consumed (CF) (in euros). The total selling price of a carcass is estimated by multiplying the market price of chicken meat for 2023 by the average carcass weight. The utilization of CBR within the framework of a cost–benefit analysis was employed to succinctly assess the economic worth of substituting soybean meal with local ingredients. The CBR was determined by dividing the SP by the CF. A CBR value above 1 indicated that the benefits derived from production surpassed the associated production costs, whereas a value below 1 suggested the opposite. The RoI is a metric used to evaluate the profitability of an investment by comparing the gain or loss achieved to the amount of money initially invested. The RoI was calculated using the following formula: RoI = (GPM/CF) × 100. The greater the value of the RoI, the more favorable the financial gains of the project being evaluated [38].

### 2.5. Statistical Analyses

Several models were used to analyze the data. A two-way factorial model including diet, strain, and their interaction was used to analyze performance and blood parameters. The experimental unit for the body and organ weights, blood parameters, tonic immobility, footpad, and feather cleanliness was individual birds. A pen was the experimental unit for feed consumption and feed conversion. The responses of the strains to the diets were similar for the performance parameters but differed for the blood parameters. Our purpose was to estimate the diet effect for each strain; therefore, all the parameters were also analyzed separately. The Shapiro–Wilk test was used to assess the normality of distribution before the analysis. When there are significant interactions, the means were separated using Student’s *t*-test. *p* values < 0.05 indicate statistical significance.

A nonparametric Kruskal–Wallis test was used for the statistical analysis of tonic immobility duration and induction numbers. The footpad dermatitis and feather cleanliness scores at the slaughter age for each strain were analyzed with a chi-square test of independence. For the footpad dermatitis scoring, there were very few birds with a score of 1 (moderate) among the Cobb strain; therefore, scores 1 and 2 were pooled, resulting in a binomial scaling for intact and severely (score 1 + 2) affected birds. For the plumage cleanliness of the Cobb broilers, there were no birds with a score of 3, and there were only a few birds that scored 2. Therefore, the data were pooled as a one–zero scaling, as 0 indicated clean plumage and 1 indicated slight soiling. In the local line, there were no birds who scored 0 (clean) or 3 (severe soiling). Therefore, all the birds from the local strain showed slight (1) or moderate (2) discoloration. When the chi-square test indicated a statistically significant effect, the cell chi-squares and the adjusted standardized residuals for each cell in the contingency table were calculated for a post hoc evaluation. The cells associated with adjusted standardized residuals greater than ±2 indicated a significant contribution of that specific cell to the omnibus chi-square [39].

## 3. Results

### 3.1. Performance of the Birds

The analyses of variance (significance of *p*-values) for body weight, feed consumption, and feed conversion ratio by diet and strain and the related means are presented in Table 3 and Table 4, respectively.

On day 10, the birds from both experiments who had been fed the SPR + BSF diet had the heaviest body weight, whereas the body weights of the birds who had been fed the SPR diet were similar to the control. The effect of the diets on the body weights of the birds on day 25 and at the slaughter age was not significant. The feed consumption of the local and Cobb birds fed the SPR + BSF diet increased from day 0 to day 10 compared to the feed consumption of those fed the SPR and control diets. The feed consumption of the chickens from both experiments was similar from days 11 to 25 and from day 26 to the slaughter age (Table 4). The total feed consumption was 5162, 5214, and 5133 g for the local chickens and 3769, 3960, and 3948 g for the Cobb chickens who had been fed the control, SPR, and SPR + BSF diets. From day 11 to day 25, the feed conversion ratio of the Cobb chickens was impaired with the SPR + BSF diet. There was no significant effect of the diets on the feed conversion ratio from day 0 to day 10, from day 25 to the slaughter age, or from day 0 to the slaughter age (Table 4).

The commercial broilers had a higher breast yield compared to the local chickens. The SPR and SPR + BSF diets significantly increased the breast yield of the local line. The diets did not affect the breast and leg yield of the Cobb broilers (Table 5 and Table 6).

### 3.2. Blood Parameters and Organ Weights

The relative weights of the liver and bursa of Fabricius were higher in the commercial strain than in the local chickens. The SPR and SPR + BSF diets reduced the relative liver weights in both strains. The relative weights of the intestine, spleen, and bursa of Fabricius were not influenced by the diets (Table 5 and Table 6).

Table 7 and Table 8 present analyses of variance (significance of *p*-values) and means for the blood parameters.

At the slaughter age, all the blood parameters measured, except for corticosterone, were higher in the commercial strain than in the local chickens. The local birds who had been fed the SPR + BSF diets had the lowest glucose levels, followed by those who had been fed the SPR and control diets (Table 8). The SPR diet increased the blood ALT levels of the local chickens. The blood AST, GGT, protein, triglycerides, and cholesterol levels of the local chickens were reduced by the SPR + BSF diet, whereas the birds who had been fed the SPR and control diets had similar levels. The diets had no significant effect on the blood glucose, AST, triglycerides, total cholesterol, uric acid, and Ca levels of the Cobb broilers (Table 8). The SPR diet tended to reduce the blood ALT levels and significantly increase the blood GGT levels of the Cobb broilers. While the blood total protein levels of the Cobb broilers decreased in the birds who had been fed the SPR + BSF diet, similar blood protein levels were obtained in the birds who had been fed the control and SPR diets. The highest creatinine levels were observed for the Cobb birds who had been fed the control diet. The diets had no significant effect on the blood corticosterone and blood uric acid levels of the chickens from either strain (Table 8).

The blood Mg and P levels of the birds from both strains who had been fed the SPR + BSF diet were reduced in the birds who had been fed the SPR + BSF diet. The lowest P level was observed for the birds who had been fed the SPR + BSF diet (Table 8). The blood Ca levels of the local chickens were affected by the diets, and the SPR + BSF diet reduced the Ca levels, but this effect was not significant in the Cobb chickens.

### 3.3. Welfare Traits of the Birds

The effect of the diet on tonic immobility duration and induction numbers for each strain is presented in Table 9. The diets did not affect tonic immobility duration and the number of inductions in either strain.

The severity of footpad dermatitis was independent of the diets for both strains (*p* > 0.05). The distribution of the footpad scores of the chickens at the slaughter age is presented in Table 10. The percentage of birds with a score of “0” was 68.95% in the local strain and 78.48% in the Cobb one. The percentage of mildly affected birds was 19.35% in the local strain and ranged between 29.17 and 39.58% within the dietary groups. The percentage of birds with a score of “2” was 11.69%. For the commercial strain, footpad dermatitis incidence was 21.52% and ranged between 27.08 and 43.75% within the diets.

The distribution of the feather cleanliness scores among the diets for each strain is presented in Table 11.

In the local strain, there were no birds who scored “0” (clean) or “3” (severe dirtiness). All the birds showed slight (score 1) or moderate (score 2) discoloration and, thus, impaired cleanliness at the slaughter age. Of the birds examined, 75.61% had a score of “1”, which indicated slight discoloration, while the percentage of birds with moderate soiling (score “2”) was 24.39%. A chi-square analysis revealed that the feather cleanliness scores of the birds did not depend on the diets in the local strain. However, in the commercial strain, the feather cleanliness scores were dependent on the diet (χ^2^ = 15.954; *p* = 0.0003; df: 2). The percentage of clean birds at the slaughter age for the commercial strain was 20.9%. The lowest (%13.4) and the highest (54.35%) number of birds with a score of “0” were found in the birds who had been fed the SPR and SPR + BSF diets, respectively. The higher cell χ^2^ values of the SPR and SPR + BSF diets by score “0” combinations revealed that the SPR diet reduced the number of clean birds, which was indicated by a greater adjusted standardized residual value (>−2). However, higher than expected numbers of clean birds were found in those who had been fed the SPR + BSF diet, with the highest contribution to the total χ^2^ and a higher adjusted standardized residual (>2).

### 3.4. Economic Performance

Table 12 and Table 13 display the statistical analysis and mean values of the GPM, CBR, and RoI estimations by diet and strain.

While the cost of the diet was about the same for the control and SPR diets, the SPR + BSF diet was 25.3% more expensive than the control diet (Table 13).

The cost of the feed consumed ranged from 2.30 to 2.87 euro/bird and from 1.70 to 2.25 euro/bird for the local and commercial strains, respectively. The total cost of the feed consumed for the control, SPR, and SPR + BSF diets was 26.0, 25.3, and 21.3% lower in the commercial strain compared to the local strain. The GPM, CBR, and RoI values achieved for the birds who had been fed the control and SPR diets were similar, higher than those of the birds who had been fed the SPR + BSF larvae diet (Table 13). Despite the current price of BSF larvae being too high to be an economically competitive ingredient, the observed CBR for the SPR + BSF diet appears to be higher than one. On the other hand, all the economic indicators (GPM, CPR, and RoI) are deteriorated significantly by the addition of 5% BSF larvae in the SPR diet.

## 4. Discussion

Soybeans are the main source of protein in broiler diets. However, soybean cultivation depends on extensive land use, leading to deforestation in Brazil and Argentina [40,41]. This has stimulated interest in nonconventional feedstuffs, such as agro-industrial by-products and BSF larvae, that offer an alternative source of protein in broiler nutrition and an opportunity for environmental sustainability. High-protein sunflower meal, brewers’ dried grain, and wheat middlings are valuable protein sources for broiler diets with an average protein content of 35, 20, and 18%, respectively [12,42,43]. BSF larvae, rich in protein (37–63%), are a promising feedstuff for replacing soybeans in broiler diets and do not compete with human nutrition [16]. Although it has been shown that agro-industrial by-products and BSF larvae can be used in broiler diets without affecting broiler performance, to our knowledge, there are no studies using a combination of them. Therefore, this study aimed to investigate the effect of diets in which soybean meals had been partially replaced with sunflower meal, brewers’ dried grain, wheat middlings, and BSF larvae meal on the performance, blood parameters, and welfare traits of commercial and local broiler chickens and evaluate the economic impacts of using alternative feedstuffs.

### 4.1. Performance of the Birds

In the present study, the body weights of the day-old chicks were similar among the groups, indicating that the chicks were distributed randomly. The average slaughter weight and feed conversion ratio were 2247 and 2.33 for the local and 2196 g 1.77 for the commercial chickens at 55 and 40 d of age, respectively. The results obtained for the Cobb chickens were lower than the body weight and the feed values specified for Cobb chickens at 40 days [43]. This was because they were fed diets containing less protein than the dietary protein values recommended for Cobb500 (21–22, 19–20, 18–19, and 17–18% of protein for starter, grower 1, grower 2, and finisher diets) [44].

Considering the growth and feed conversion ratio, brewers’ dried grain can be included in a diet in concentrations by up to 20% [7,14,45]. Ramo Rao et al. [4] found that the replacement of soybean with sunflower meal up to 67% in the starter and 100% in the finisher diets increased feed consumption and discouraged feed efficiency. On the other hand, the replacement of soybean meal with high-protein sunflower meal by up to 50% using multi-enzyme mixtures did not affect the growth performance of broilers [9]. In our study, the SPR diet did not affect the body weight, feed consumption, and feed conversion rate of broilers. This result may be related to the use of enzymes and the lower rates of soybean replacement in our study.

The effect of BSF larvae meal inclusion in the broiler diets on body weight, feed consumption, and feed conversion may vary due to the level of inclusion [24,46,47], larvae form (live, dried, full-fat, or de-fatted) [46,48,49], and BSF larvae composition, which can be modified by the diet of the larvae [17,50]. The soybean-based diets replaced with BSF larvae by up to 50 and 100% reduced the body weight of Ross broilers [24]; thus, low levels would be more suitable [21]. The higher body weight obtained on day 10 in the chicks who had been fed the SPR + BSF diet showed that a 5% substitution of soybean with BSF larvae meal the addition of 5% BSF larval meal to the SPR diet improved the early growth performance of chicks from both strains. This result was similar to the findings reported by Dabbou et al. [21]. It is possible to attribute this result to two reasons, as follows: (1) The occurrence of initial colonization of the intestinal microbiota has an impact on growth performance and daily gain [51]. BSF larvae meal may have affected the weight gain in this period by contributing to the formation of intestinal microbiota in the chicks. (2) The increase in body weight was accompanied by the increased feed consumption and appetite of the chicks during the same period. Naturally, chickens like to consume larvae [52]. Heuel et al. [53] reported that diets containing BSF larvae meal were slightly superior in palatability to a diet including soybean meal, which may increase feed consumption. However, this effect disappeared at later ages; birds from different dietary groups measured similar body weights at day 25 and at the slaughter age. An impaired feed conversion was obtained for the commercial birds consuming the SPR + BSF diet from day 11 to day 25; however, the diets did not affect the final feed conversion of the local and commercial birds. These results showed that SPR could be used as an alternative diet to soybean-based diets without affecting broiler performance and that adding 5% of BSF to the SPR diet would not affect growth performance and the feed conversion ratio.

A concentration of Brewers’ dried grain up to 20% increased the eviscerated yield but reduced the drumstick yield [13]. Araujo et al. [54] reported that the inclusion of sunflower meal with enzymes in concentrations up to 24% in broiler diets from day 21 to day 42 reduced carcass parameters. Murawska et al. [24] noted that replacing soybeans with full-fat BSF larvae by up to 100% reduced breast yield. Since soybeans were substituted with three feedstuffs in our study, it is difficult to directly compare them with these studies. In the present study, although the breast yield of the local chickens increased with the SPR and SPR + BSF diets, this effect was not significant for the commercial broilers. This difference between the response of breast yield to the diets in the commercial and the local strain may be related to the chickens’ growth rate and carcass composition.

### 4.2. Blood Parameters and Organ Weights

Bongiorno et al. [48] reported that live BSF larvae increased the weights of the spleen and bursa of Fabricius and reduced GGT activity in the blood indicating positive effects on liver health and the immune system. Schiavone et al. [55] also showed that BSF larvae fat would not affect the liver health of commercial broilers, confirming its nutritional adequacy. The liver is a multi-purpose supportive organ of the digestive system and plays a role in digestion, with an involvement in lipid, carbohydrate, and protein metabolism, and glucose homeostasis to meet energy demands [56]. Glycolysis is the main pathway for glucose catabolism, which depends on the activities of enzymes [57]. The lower liver and blood glucose levels obtained in the local broilers who had been fed the SPR + BSF diet indicated that glycolysis was inhibited. Similarly, Chen et al. [58] reported that the glucose metabolism changed in shrimps who had been fed BSF larvae meal. However, this effect was not observed in the commercial broilers in our study. The present study showed that the blood parameters of the two strains differed in their response to the diets: i.e., the diets did not affect the blood glucose, ALT, AST, triglyceride, cholesterol, and uric acid levels of the commercial chickens. This result might be due to the feeding period, which was longer for the local chickens compared to the commercial chickens, and/or their different growth rates and genetic backgrounds.

In the literature, the results on the effects of by-products and BSF larvae on the blood profile of chickens are inconsistent and depend on the supplementation level. Parpinelli et al. [59] indicated that the inclusion of up to 8% brewers’ dried grain did not affect the triglyceride, uric acid, and creatinine levels; however, it increased the ALT and AST levels of broilers. An increase in the blood ALT and AST levels due to the inclusion of BSF larvae meal was also reported in [47,60]. In contrast to these findings, Dabbou et al. [49] and Attia et al. [61] reported that a soybean-based diet partially substituted with BSF larvae did not affect the serum ALT, AST, triglycerides, or uric acid levels. ALT and AST are specific indicators of liver health. The increased blood ALT levels of the local chickens who had been fed the SPR diet indicated liver cell damage. However, this situation was thought to be tolerable because the mortality rates were very low for the dietary groups. On the other hand, the decreased blood ALT levels with the inclusion of BSF in the SPR diets of local chickens may indicate a normal liver metabolism, which was accompanied by lower GGT levels. These results indicated that adding BSF to the SPR diet protected the liver of the local chickens. The blood cholesterol and triglyceride levels also decreased in the local chickens, agreeing with Bongiorno et al. [48]. This result may be related to the level of chitin in the BSF diet, which reduces lipid absorption in the intestine by binding lipids or fatty acids [61]. In our study, the chitin level was found to be 8.32% and was consistent with Marono et al. [36]. On the other hand, the creatinine levels of local chickens were not influenced by the diets, showing that the diets did not significantly affect renal functions, while the creatinine levels in the commercial chickens decreased in the diets implementing alternative feedstuffs.

There are conflicting results related to the effect of insect larvae on blood minerals. Loponte et al. [62] found no effect of BSF larvae on blood minerals when 25 and 50% of soybean meal were substituted with larvae protein. Gariglio et al. [25] reported that the inclusion of 0 to 9% of partially de-fatted BSF larvae meal in Muscovy ducks’ diets did not affect the levels of blood Ca and P, while the blood Mg levels decreased linearly. In our study, the blood Mg and P levels decreased with BSF larvae meal inclusion in both strains, which agreed with Bovera et al.’s study [63]. The response of the strains to the BSF larvae meal-supplemented diet differed in the blood Ca levels; the BSF larvae diet reduced the blood Ca content of the local chickens. In contrast to this finding, Marono et al. [64] reported higher blood Ca levels in laying hens who had been fed BSL larvae meal.

### 4.3. Welfare Traits of the Birds

In this study, tonic immobility test responses, footpad dermatitis, and plumage cleanliness were measured to evaluate the effect of SPR and SPR + BSF diets on the welfare of broilers from local and commercial strains. There was not any significant effect on the tonic immobility responses. Ipema et al. [65] reported a reduced tonic immobility duration and improved activity, but no further improvement in health-related welfare traits, such as lameness and feather cleanliness, were observed when live BSF larvae were administered with a replacement level of either 5% or 10% of the dietary intake.

The plumage cleanliness score was the only welfare measure that was significantly affected by the diet in the commercial broilers. Our results revealed that the SPR diet had a negative impact on the plumage score in the commercial strain. However, the inclusion of BSF larvae meal into the SPR diet seemed to alleviate this negative effect. Footpad health and other welfare indicators including plumage cleanliness, breast irritation, hock burn, and gait were negatively related to litter quality [66]. The positive effect of BSF inclusion into the diet might have been related to the health of the digestive system and microbiota composition, which directly affect litter quality. Indeed, BSF larvae meal utilization has been shown to positively influence cecal microbiota and the gut’s mucin dynamics, as indicated by an increased level of beneficial bacterial population, particularly lactic acid, and an increase in villi mucins [67,68]. Therefore, we may speculate that the inclusion of BSF larvae at a concentration of 5% into the SPR diet accounted for better plumage conditions. Another possible explanation for the improved cleanliness score in the broilers who had been fed the SPR + BSF diet could be the reduced serum Mg and P levels of the birds from both strains. This may modify the moisture content of the excreta and, thus, plumage cleanliness, as increasing dietary Mg levels have been reported to increase excreta moisture in broilers [69]. The cleanliness of the local broilers was not influenced by the SPR and SPR + BSF treatments. However, 100% of the birds presented either slight or moderate soiling, and this might be associated to the overall higher moisture content of the excreta from the local strain. Indeed, excreta dry matter was affected by strain, being higher in the Cobb chickens (30.04 ± 1.35%) than in the local (25.99 ± 0.89%) strain in this study (authors’ unpublished data).

In our study, the footpad dermatitis scores were not affected by diet. Contrary to our findings, positive effects on the footpad health and hock burn conditions of broilers have been reported after the inclusion of BSF larvae meal into the diet or the administration of live larvae separately from the diet [65]. The differences between the two studies might be related to the inclusion level of BSF larvae, which was 5% of the diet in our case and 8% of the diet’s dry matter in their study. The differences in the inclusion levels of BSF larvae might have modified the magnitude of their effect on the footpad dermatitis scores of the broilers from both strains.

### 4.4. Economic Performance

The economic performance was assessed utilizing key indicators such as GPM, CBR, and RoI. Beyond chicken performance, the economy of production forms the basis for the inclusion of alternative feedstuffs as a protein source into chicken diets to partly substitute soybean meal in the sector. The price of feedstuffs has a great impact on diet cost and on the profitability of production. The findings of our economic analysis indicate that the SPR diet had similar economic results to the control diet. However, adding BSF larvae meal into the SPR diet resulted in a negative economic effect on chicken meat production. Nevertheless, the CBR of this particular diet demonstrated a relatively lower value compared to both the control and SPR diets. Contrary to our findings, Onsongo et al. [38] reported that the inclusion of BSF larvae meal as a substitute for soybean meal in chicken diets resulted in 16 and 25% higher CBR and RoI, respectively, compared to the control diet, the unit price of which was 19% more expensive. This effect was observed when the BSF larvae meal constituted 42.0% and 55.5% of the crude protein in the starter and finisher diets, respectively. The controversial state of the findings in our research can be attributed to the high prices for BSF in Türkiye, due to the low availability of this resource and, therefore, the small scale of production. The cost of the larvae also surged in our study due to the usage of dried larvae. Using live larvae, produced within one’s own company, will likely reduce the costs. Another issue is that the cost of production with a local strain is high, which is an expected result. Increasing profitability can be achieved by modifying the selling prices of local products.

## 5. Conclusions

The availability of alternative protein sources that can partly replace soybean at acceptable prices is important for sustainable poultry meat production. The findings of this study indicated that the SPR diet, in which soybean meal was partially substituted with agro-industrial by-products, met the nutritional requirements of chickens without negatively affecting the slaughter weight and feed conversion of both local and commercial chickens. The incorporation of BSF larvae meal into the SPR diet improved the growth of the chicken during the first 10 d post hatching. Furthermore, the SPR + BSF diet seemed to alleviate the negative impact of the SPR diet on the plumage score of the commercial broilers at the slaughter age. The local strain, which had a longer production period of 55 d, exhibited compromised cleanliness scores and plumage conditions across all individuals, irrespective of their diet. This response may be related to differences in the diet response and gut health between strains, as gut health is associated with plumage cleanliness. Future research should explore how the microbiota is affected in chickens who are fed SPR and SPR + BSF diets. On the other hand, the economic analysis in this study showed that the SPR + BSF diet had a negative impact on the GPM, CBS, and RoI. The profitability of broiler production largely depends on the price of BSF larvae meal. In future studies, economic sustainability will be better assessed by making cost estimates under various price scenarios.

## Figures and Tables

**Table 1 animals-14-00314-t001:** Ingredients and nutrient composition of the experimental diets for the starter (0–10 d), grower (11–25 d), and finisher (26-slaughter age ^1^) periods.

	Control			SPR			SPR + BSF		
	Starter	Grower	Finisher	Starter	Grower	Finisher	Starter	Grower	Finisher
Corn	45.28	51.24	57.34	39.18	44.44	47.44	41.3	46.54	49.64
Wheat	11.86	14.86	15.00	12.5.0	14.50	15.50	12.18	14.50	15.50
Soybean meal	34.33	27.90	23.20	29.8	21.10	14.6	25.10	16.30	9.70
Sunflower meal	-	-	-	3.58	6.30	8.00	3.63	6.30	8.00
Brewers’ dried grain	-	-	-	2.58	3.08	4.00	2.63	3.08	4.00
Wheat middling	-	-	-	2.58	3.08	4.00	2.63	3.08	4.00
BSF larvae	-	-	-	-	-	-	5.00	5.00	5.00
Sunflower oil	5.88	4.00	3.00	7.13	5.5.0	5.00	4.88	3.20	2.70
Limestone	0.50	0.30	0.20	0.50	0.30	0.20	0.50	0.30	0.20
DCP	1.00	0.80	0.60	1.00	0.80	0.60	1.00	0.80	0.60
Vit+ min premix ^2^	0.25	0.25	0.25	0.25	0.25	0.25	0.25	0.25	0.25
NaCl	0.20	0.20	0.20	0.20	0.20	0.20	0.20	0.20	0.20
Lysine (HCL—% 78)	0.50	0.30	0.15	0.50	0.30	0.15	0.50	0.30	0.15
Methionine dl (% 99)	0.10	0.05	0.01	0.1	0.05	0.01	0.01	0.05	0.01
Threonine	0.05	0.05	-	0.05	0.05	-	0.05	0.05	-
Enzyme ^3^	0.05	0.05	0.05	0.05	0.05	0.05	0.05	0.05	0.05
Analyzed Nutrient Composition ^4^									
ME kcal/kg diet	2984	2923	2904	2992	2921	2904	2991	2919	2903
CP, %	20.78	18.68	17.00	20.74	18.65	17.05	20.78	18.64	17.04
EE, %	8.49	6.63	5.79	9.41	8.11	7.52	9.38	7.72	7.52
CF, %	2.91	2.61	2.35	3.70	3.69	3.77	3.94	3.92	3.99
Ca, %	1.08	1.04	1.02	1.08	1.03	0.99	1.13	1.08	1.05
Total P	0.50	0.44	0.38	0.55	0.52	0.47	0.55	0.52	0.48

^1^ Slaughter age was 55 and 40 d for the local and commercial strains, respectively. ^2^ Vitamin + mineral premix provided per 2.5 kg feed of diet: Vitamin A, 15,000,000 IU; Vitamin D3, 3,000,000 IU; Vitamin E, 50,000 mg; Vitamin K3, 4000 mg; Vitamin B1, 3000 mg; Vitamin B2, 6000 mg; Niacinamid, 40,000 mg; Vitamin B6, 5000 mg; Vitamin B12, 30 mg; Calcium-D-Pantothenate, 15.000 mg; Biotin, 75 mg; Folic acid, 1000 mg; Choline Chloride, 400,000 mg; Manganese, 80.000 mg; Iron, 60.000 mg; Copper, 5000 mg; Zinc, 60.000 mg; Iodine, 2.000 mg; and Selenium, 150 mg. ^3^ Rovabio (50 gr) + Natuphos E (100 gr) BASF. ^4^ ME: metabolizable energy; CP: crude protein; EE: ether extract; CF: crude fiber; Ca: calcium; Total P: total phosphorus.

**Table 2 animals-14-00314-t002:** Analyzed nutrient compositions of alternative feedstuffs (% on dry matter).

	Sunflower Meal	Brewers’ Dried Grain	Wheat Middlings	BSF Larvae
Metabolizable energy, kcal/kg	2108	1565	1837	5381
Dry matter	90.46	90.53	88.20	95.52
Crude protein	41.78	28.61	17.53	42.62
Ether extract	1.65	2.68	4.34	42.54
Crude fiber	14.37	20.20	7.94	11.04
Crude ash	6.92	4.11	5.98	6.29
Starch	-	2.71	17.87	-
Total sugar	8.18	2.07	-	1.79
Neutral detergent fiber	37.57	74.68	32.59	17.56
Acid detergent fiber	29.88	28.97	10.78	13.36
Acid detergent insoluble nitrogen, %	-	-	-	0.74
Methionine	0.70	0.24	0.26	0.65
Lysine	1.59	0.88	0.68	1.35
Threonine	1.91	1.14	0.56	2.73
Calcium	0.43	2.55	1.05	1.70
Total phosphorus	1.38	0.78	0.14	0.62
Fatty acids				
Σ SFA ^1^, g/100 g lipid	29.45	27.43	19.21	71.89
Σ MUFA ^2^, g/100 g lipid	27.96	18.79	17.56	17.09
Σ PUFA ^3^, g/100 g lipid	42.59	53.78	62.89	11.02

^1^ SFA: saturated fatty acids; ^2^ MUFA: monounsaturated fatty acids; ^3^ PUFA: polyunsaturated fatty acids.

**Table 3 animals-14-00314-t003:** Analyses of variance (significance of *p*-values) for body weight, feed consumption, and feed conversion ratio by stain and diet.

		Strain	Diet	Strain × Diet
Body weight	0 d	<0.001	0.298	0.279
	10 d	<0.001	<0.001	0.348
	25 d	<0.001	0.806	0.565
	Slaughter age	0.317	0.134	0.672
Feed consumption	0–10 d	0.167	<0.001	0.816
	11–25 d	<0.001	0.530	0.567
	26 d—Slaughter age	<0.001	0.530	0.567
	Total	<0.001	0.493	0.586
Feed conversion	0–10 d	<0.001	0.128	0.667
	11–25 d	<0.001	0.276	0.388
	26 d—Slaughter age	<0.001	0.864	0.674
	Total	<0.001	0.908	0.688

**Table 4 animals-14-00314-t004:** Effects of the diets ^1^ on body weight, feed consumption, and feed conversion ratio in local and commercial strains.

Performance		Local Strain					Commercial Strain				
		Control	SPR	SPR + BSF	SEM ^2^	*p*-Value	Control	SPR	SPR + BSF	SEM	*p*-Value
Body weight	0 d	35.32	35.37	35.34	0.27	0.993	36.96	36.63	36.38	0.29	0.368
	10 d	218 ^b^	213 ^b^	230 ^a^	3.1	<0.001	236 ^b^	242 ^ab^	251 ^a^	3.4	0.012
	25 d	792	777	805	12.6	0.280	998	1007	989	15.2	0.692
	Slaughter age ^3^	2223	2283	2236	30.7	0.353	2161	2247	2181	31.3	0.112
Feed consumption	0–10 d	366 ^b^	390 ^b^	444 ^a^	15.6	0.008	356	372	414	18.6	0.106
	11–25 d	975	968	985	19.8	0.844	1120	1150	1161	19.4	0.344
	25 d—slaughter age	3820	3855	3703	96.4	0.519	2292	2438	2372	109.0	0.645
	Total	5162	5214	5133	101.4	0.849	3769	3960	3948	118.1	0.458
Feed conversion	0–10 d	2.03	2.21	2.28	0.09	0.191	1.79	1.81	1.91	0.092	0.629
	11–25 d	1.71	1.68	1.70	0.042	0.918	1.48 ^b^	1.49 ^b^	1.57 ^a^	0.020	0.021
	25 d—slaughter age	2.65	2.58	2.60	0.064	0.736	1.93	1.96	1.92	0.086	0.932
	Total	2.35	2.32	2.33	0.0336	0.844	1.76	1.78	1.79	0.052	0.869

^a,b^ The means in the same row within a strain with different superscripts differ significantly (*p* < 0.05). ^1^ Diets: control: corn–soybean-based diet; SPR: the soybean in the control diet was partially replaced with sunflower meal, brewers’ dried grain, and wheat middlings; SPR + BSF: black soldier fly dried larvae were added to the SPR diet. ^2^ SEM: standard error of means. ^3^ Slaughter age was 55 and 40 d for the local and commercial strains, respectively.

**Table 5 animals-14-00314-t005:** Analyses of variance (significance of *p*-values) for the relative weights of breast, leg, liver, intestine, spleen, and bursa of Fabricius by strain and diet.

	Strain	Diet	Strain × Diet
Breast	<0.001	0.017	0.027
Leg	0.215	0.651	0.974
Liver	<0.001	0.147	0.391
Intestine	0.583	0.016	0.064
Spleen	0.374	0.342	0.953
Bursa of Fabricius	<0.001	0.199	0.261

**Table 6 animals-14-00314-t006:** Effects of the diets ^1^ on the relative weights of breast, leg, digestive, and immune system organs in local and commercial strains at the slaughter age ^2^.

	Local Strain					Commercial Strain			
	Control	SPR	SPR + BSF	SEM ^3^	*p*-Value ^4^	SPR	SPR + BSF	SEM	*p*-Value
Breast	19.54 ^b^	20.70 ^a^	21.30 ^a^	0.399	0.016	23.08	23.69	0.370	0.518
Leg	18.90	18.69	18.97	0.268	0.744	19.01	19.19	0.238	0.867
Liver	1.62 ^a^	1.49 ^b^	1.52 ^b^	0.043	0.007	1.48	1.46	0.033	0.853
Intestine	4.25	4.62	4.09	0.17	0.122	3.64	3.71	0.126	0.891
Spleen	0.112	0.116	0.112	0.0064	0.922	0.099	0.083	0.0055	0.104
Bursa of Fabricius	0.0516	0.0467	0.0506	0.04177	0.701	0.0484	0.0543	0.00442	0.476

^a,b^ The means in the same row within a strain with different superscripts differ significantly (*p* < 0.05). ^1^ Diets: control: corn–soybean-based diet; SPR: the soybean in the control diet was partially replaced with sunflower meal, brewer’s dried grain, and wheat middlings; SPR + BSF: black soldier fly dried larvae were added to the SPR diet. ^2^ Slaughter age was 55 and 40 d for the local and commercial strains, respectively. ^3^ SEM: standard error of means. ^4^
*p*-Value: Significance

**Table 7 animals-14-00314-t007:** Analyses of variance (significance of *p*-values) for blood parameters by diet and strain.

Blood Parameters ^1^	Strain	Diet	Strain × Diet
Corticosterone	0.926	0.926	0.971
Glucose	<0.001	0.007	<0.001
ALT	<0.001	0.518	<0.001
AST	<0.001	0.030	0.048
GGT	<0.001	0.030	0.021
Protein	<0.001	<0.001	0.004
Triglycerides	<0.001	0.358	0.034
Cholesterol	<0.001	<0.001	0.073
Creatinine	<0.001	0.041	0.094
Uric acid	0.003	0.038	0.281
Mg	<0.001	<0.001	<0.001
Ca	<0.001	<0.001	<0.001
P	<0.001	<0.001	0.012

^1^ ALT: alanine aminotransferase; AST: aspartate aminotransferase; GGT: gamma-glutamyl transferase.

**Table 8 animals-14-00314-t008:** Effects of the diets ^1^ on the blood corticosterone and blood metabolites in local and commercial strains at slaughter age ^2^.

Blood Metabolites ^4^	Local Strain					Commercial Strain				
	Control	SPR	SPR + BSF	SEM ^3^	*p*-Value	Control	SPR	SPR + BSF	SEM	*p*-Value
Corticosterone, ng/m	8.1	7.44	8.18	0.592	0.222	8.77	7.59	8.31	0.740	0.531
Glucose, mg/dL	183 ^a^	159 ^b^	110 ^c^	9.5	<0.001	212	226	221	7.0	0.354
ALT, U/L	1.57 ^b^	2.47 ^a^	1.40 ^b^	0.252	0.015	3.17	2.54	3.18	0.288	0.062
AST, U/L	239 ^a^	231 ^a^	146 ^b^	13.3	<0.001	411	396	386	28.1	0.837
GGT, U/L	15.82^a^	15.05 ^a^	9.21 ^b^	1.107	<0.001	18.39 ^b^	20.75 ^a^	17.44 ^b^	0.803	0.017
Protein, g/L	22.75 ^a^	24.11^a^	14.12 ^b^	1.463	<0.001	28.32 ^a^	29.49 ^a^	26.28 ^b^	0.555	<0.001
Triglyceride, mg/dL	14.51 ^a^	14.06 ^a^	8.55 ^b^	1.402	0.008	22.39	21.93	23.93	1.825	0.728
Cholesterol, mg/dL	88.39 ^a^	79.82 ^a^	55.43 ^b^	5.507	<0.001	133	134	126	4.1	0.304
Creatinine, mg/dL	0.183	0.197	0.153	0.0163	0.175	0.259 ^a^	0.224 ^b^	0.221 ^b^	0.0111	0.044
Uric acid, mg/dL	1.71	1.64	1.17	0.187	0.094	2.24	1.96	1.94	0.158	0.365
Mg, mg/dL	2.32 ^a^	2.06 ^a^	1.32 ^b^	0.114	<0.001	2.52 ^ab^	2.74 ^a^	2.43 ^b^	0.089	0.051
Ca, mg/dL	7.80 ^a^	7.55 ^a^	4.92 ^b^	0.415	<0.001	8.33	8.66	8.22	0.169	0.187
P, mg/dL	6.53 ^a^	5.85 ^a^	4.06 ^b^	0.323	<0.001	7.49 ^a^	7.50 ^a^	6.61 ^b^	0.171	<0.001

^a,b^ The means in the same row within a strain with different superscripts differ significantly (*p* < 0.05). ^1^ Diets: control: corn–soybean-based diet; SPR: the soybean in the control diet was partially replaced with sunflower meal, brewer’s dried grain, and wheat middlings; SPR + BSF: black soldier fly dried larvae were added to the SPR diet. ^2^ Slaughter age was 55 and 40 d for the local and commercial strains, respectively. ^3^ SEM: standard error of means. ^4^ Alanine aminotransferase (ALT), aspartate aminotransferase (AST), and gamma-glutamyl transferase (GGT).

**Table 9 animals-14-00314-t009:** The effect of the diets ^1^ on tonic immobility (TI) test responses of local and commercial strains on day 28 and at the slaughter age ^2^.

		Local Strain					Commercial Strain				
		Control	SPR	SPR + BSF	SEM ^3^	*p*-Value	Control	SPR	SPR + BSF	SEM	*p*-Value
D28	TI duration (s)	68.50	52.83	63.08	12.7	0.594	81.67	89.67	98.83	9.20	0.375
	Induction number	2.17	1.83	1.58	0.26	0.249	1.08	1.33	1.42	0.15	0.346
Slaughter age	TI duration (s)	79.25	81.08	79.08	12.51	0.977	104.08	97.00	87.67	10.57	0.441
	Induction number	2.08	1.83	1.67	0.24	0.489	1.42	1.17	1.17	0.15	0.738

^1^ Diets: control: corn–soybean-based diet; SPR: the soybean in the control diet was partially replaced with sunflower meal, brewers’ dried grain, and wheat middlings; SPR + BSF: black soldier fly dried larvae were added to the SPR diet. ^2^ Slaughter age was 55 and 40 d for the local and commercial strains, respectively. ^3^ SEM: standard error of means.

**Table 10 animals-14-00314-t010:** Distribution (%) of the footpad dermatitis scores of chickens from local and commercial strains by the diets ^1^ at slaughter age ^2^ (observed numbers are given in parenthesis).

Diets	Footpad Dermatitis (%)				
	Local Strain			Commercial Strain	
Control	32.75 (*n* = 56) Cell χ^2^ = 0.000	29.17 (*n* = 14) Cell χ^2^ = 0.179	37.93 (*n* = 11) Cell χ^2^ = 0.247	32.57 (*n* = 57) Cell χ^2^ = 0.289	43.75 (*n* = 21) Cell χ^2^ = 1.056
SPR	29.82 (*n* = 51) Cell χ^2^ = 0.826	39.58 (*n* = 19) Cell χ^2^ = 0.462	48.28 (*n* = 14) Cell χ^2^ = 1.176	34.86 (*n* = 61) Cell χ^2^ = 0.148	27.08 (*n* = 13) Cell χ^2^ = 0.53
SPR + BSF	37.43 (*n* = 64) Cell χ^2^ = 0.800	31.25 (*n* = 15) Cell χ^2^ = 0.070	13.75 (*n* = 4) Cell χ^2^ = 3.354	32.57 (*n* = 57) Cell χ^2^ = 0.029	29.17 (*n* = 14) Cell χ^2^ = 0.107

^1^ Diets: control: corn–soybean-based diet; SPR: the soybean in the control diet was partially replaced with sunflower meal, brewers’ dried grain, and wheat middlings; SPR + BSF: black soldier fly dried larvae were added to the SPR diet. ^2^ Slaughter age was 55 and 40 d for the local and commercial strains, respectively.

**Table 11 animals-14-00314-t011:** Distribution (%) of the feather cleanliness scores of chickens from local and commercial strains by the diets ^1^ at slaughter age ^2^ (observed numbers are given in parenthesis).

Diets	Local Strain		Commercial	
	1 (Slight Soiling)	2 (Moderate Soiling)	0 (Clean Plumage)	1 (Slight Soiling)
Control	35.48 (*n* = 66) Cell χ^2^ = 0.502	23.33 (*n* = 14) Cell χ^2^ = 1.557	32.61 (*n* = 15) Cell χ^2^ = 0.049	35.06 (*n* = 61) Cell χ^2^ = 0.013
SPR	32.80 (*n* = 61) Cell χ^2^ = 0.099	38.33 (*n* = 23) Cell χ^2^ = 0.308	13.04 (*n* = 6) Cell χ^2^ = 5.622	38.51 (*n* = 67) Cell χ^2^ = 1.486
SPR + BSF	31.72 (*n* = 59) Cell χ^2^ = 0.145	38.33 (*n* = 23) Cell χ^2^ = 0.450	54.35 (*n* = 25) Cell χ^2^ = 6.946	26.44 (*n* = 46) Cell χ^2^ = 1.836
Total (%)	75.61 (*n* = 186)	24.39 (*n* = 60)	20.9 (*n* = 46)	79.09 (*n* = 174)
	Pearson χ^2^ = 3.062, *p* = 0.216		Pearson χ^2^ = 15.954, *p* = 0.0003	

^1^ Diets: control: corn–soybean-based diet; SPR: the soybean in the control diet was partially replaced with sunflower meal, brewer’s dried grain, and wheat middlings; SPR + BSF: black soldier fly dried larvae were added to the SPR diet. ^2^ Slaughter age was 55 and 40 d for the local and commercial strains, respectively.

**Table 12 animals-14-00314-t012:** Analyses of variance (significance of *p*-values) for economic indicators by diet and strain.

	Strain	Diet	Strain × Diet
Cost of feed consumed	<0.001	<0.001	0.953
Total selling price	<0.001	0.408	0.219
Gross profit margin	<0.001	<0.001	0.185
Cost–benefit ratio	0.001	<0.001	0.675
Return on investment	0.001	<0.001	0.675

**Table 13 animals-14-00314-t013:** Effects of the diets ^1^ on the economic indicators ^2^ of production by strains.

Economic Indicators	Local Strain					Commercial Strain				
	Control	SPR	SPR + BSF	SEM	*p*	Control	SPR	SPR + BSF	SEM	*p*
Cost of feed (EUR /kg)	0.4460	0.4474	0.5610			0.4519	0.4532	0.5666		
CF	2.30 ^b^	2.33 ^b^	2.87 ^a^	0.043	<0.001	1.70 ^b^	1.74 ^b^	2.25 ^a^	0.043	<0.001
Selling price (EUR per kg of carcass)	2.83	2.83	2.83			2.26	2.26	2.26		
SP	4.65	4.71	4.62	0.046	0.345	3.69	3.75	3.79	0.036	0.207
GPM	2.35 ^a^	2.38 ^a^	1.74 ^b^	0.032	<0.001	1.99 ^a^	2.01 ^a^	1.54	0.060	<0.001
CBR	2.03 ^a^	2.02 ^a^	1.61 ^b^	0.026	<0.001	2.17 ^a^	2.16 ^a^	1.68 ^b^	0.054	<0.001
RoI	102.7 ^a^	102.2 ^a^	60.8 ^b^	2.62	<0.001	117.5 ^a^	115.7 ^a^	68.4 ^b^	5.43	<0.001

^a,b^ The means in the same row within a strain with different superscripts differ significantly (*p* < 0.05). ^1^ Diets: control: corn–soybean-based diet; SPR: the soybean in the control diet was partially replaced with sunflower meal, brewer’s dried grain, and wheat middlings; SPR + BSF: black soldier fly dried larvae were added to the SPR diet. ^2^ CF: cost of feed consumed (euro/bird); SP: Total selling price of carcass (euro/bird); GPM: gross profit margin; CBR: cost–benefit ratio; and RoI: return on investment.

## Data Availability

The datasets of the current study and the models used are available from the corresponding author upon reasonable request.

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
