# Peer review of "Effects of Partial Replacement of Soybean with Local Alternative Sources on Growth, Blood Parameters, Welfare, and Economic Indicators of Local and Commercial Broilers"

_animals, 2024, doi:10.3390/ani14020314_

Round 1

Reviewer 1 Report

Comments and Suggestions for Authors

Dear Authors, I received a paper for review titled Effects of partial replacement of soybean on the growth performance, blood parameters, and animal welfare and economic indicators of local and commercial broilers.

Study evaluated the effect of  partial replacement of soybean with alternative local agri-industry by-products and black soldier fly (BSF) larvae meal on broiler growth performance, blood biochemistry, welfare and subsequently economic performance.

The work analyzes fashionable soy substitutes in the diet of broiler chickens. The topic itself seems interesting and consistent with current trends in research in broiler nutrition. The experiment was planned correctly, the number of repetitions seems to ensure correct statistical analyses. The conclusions supported by the results and previous research are correct. Detailed feed analysis makes the final results credible. However, it should be noted that the observations obtained on such a small number of birds, on a test farm, may have a completely different picture (including the economic one, where the quality of sold livestock is influenced by factors that are not present in university trial experiments). 

After reading the work several times, a few general observations and suggestions come to mind.

The work requires  editing and language checking (preferably by a native speaker). Long, complex sentences, often with repetitions, make reading the text very difficult. Moreover, the title itself requires modification to avoid repetitions.

The work requires minor additions and correction to meet the requirements set for articles of this type by Animals MDPI. Some of the comments are below:

L12-16: please cut the sentence into the shorter parts 

L49-52: please, cut the sentence into the shorter parts.

L54: process. 

2.1. Cobb- what cobb? cob 500? other type?

Why these two lines were chosen? What where the criteria for choosing this type of chicken. Please, describe more the local line. For many readers, even those involved in the poultry industry, local varieties of chickens, their requirements and basic weight, rearing, etc. standards are unfamiliar.

Please, add the info about vaccination and any prophylactic program that was used- vitamins, minerals, acidifier (in water). 

What about the coccidiostats - were any of them use in the feed (which one, when?)

L427- there is no number of the reference, please, add it.

Authors noticed that the response of commercial chickens to the diets was different from that of local chickens. What may be the reason for the effect on blood biochemical parameters? two breeds present two types - slow-growing and long-growing, their metabolism and physiology are slightly different - could this have influenced the results?

What, in your opinion (and based on the previous study), is the reason of the observed effect- diets on a weight at day 10? Why, in your opinion there were no effect in older birds? You mentioned about gut health- could you described it in more details?

Comments on the Quality of English Language

Comments were described in suggestions for Authors

Author Response

We appreciate the time and effort that you have dedicated to providing valuable feedback and comments to improve the quality of the manuscript. I have carefully revised the manuscript and incorporated comments within the manuscript. Changes are highlighted in yellow.

We hope that the revision is satisfactory.

Kind regards

Servet Yalcin

My response to the reviewers' comments

Comments and Suggestions for Authors

Comment: Dear Authors, I received a paper for review titled Effects of partial replacement of soybean on the growth performance, blood parameters, and animal welfare and economic indicators of local and commercial broilers.

Study evaluated the effect of  partial replacement of soybean with alternative local agri-industry by-products and black soldier fly (BSF) larvae meal on broiler growth performance, blood biochemistry, welfare and subsequently economic performance.

The work analyzes fashionable soy substitutes in the diet of broiler chickens. The topic itself seems interesting and consistent with current trends in research in broiler nutrition. The experiment was planned correctly, the number of repetitions seems to ensure correct statistical analyses. The conclusions supported by the results and previous research are correct. Detailed feed analysis makes the final results credible. However, it should be noted that the observations obtained on such a small number of birds, on a test farm, may have a completely different picture (including the economic one, where the quality of sold livestock is influenced by factors that are not present in university trial experiments). 

Response: Thank you for giving us the opportunity to submit the revised manuscript. We appreciate for providing review comments. The economic analysis is based on performance and market prices, Therefore, we do not expect much change in economic analysis results under field conditions.

Comment: After reading the work several times, a few general observations and suggestions come to mind. The work requires  editing and language checking (preferably by a native speaker). Long, complex sentences, often with repetitions, make reading the text very difficult. Moreover, the title itself requires modification to avoid repetitions.

 Response: The manuscript has been edited. Long sentences were shortened by avoiding repetitions. Title: This paper includes the result of a comprehensive study. Articles including other parameters are being prepared. Therefore, we believe that leaving these parameters in the title will explain this article better.

Comments: The work requires minor additions and correction to meet the requirements set for articles of this type by Animals MDPI. Some of the comments are below:

 L12-16: please cut the sentence into the shorter parts:

Response: It was changed in L 11-15: “While the demand for poultry meat is increasing, arable land for crop production is limited. Therefore, the use of locally produced alternative sources in chicken diets becomes a necessity. The substitution of soybean with 1. local by-products such as sunflower meal, brewers’ dried grain, wheat middlings, or 2. a combination of local by-products with black soldier larvae meal was evaluated in broilers.”

Comments L49-52: please, cut the sentence into the shorter parts:

Response: The necessary changes were made in L 48-52. “Although their use is limited due to their comparatively higher fiber and lower lysine and methionine levels than soybean meal, this limitation can be overcome by adding enzymes into the diets. The addition of enzymes has been shown to increase the nutritional value and digestibility of non-starch polysaccharides and proteins in broilers [7, 8, 9].”

Comments L54: process.

Response: A Dot was included. 

Comments 2.1. Cobb- what cobb? cob 500? other type?

Response: It is Cobb 500. It was added.

Comments Why these two lines were chosen? What where the criteria for choosing this type of chicken. Please, describe more the local line. For many readers, even those involved in the poultry industry, local varieties of chickens, their requirements and basic weight, rearing, etc. standards are unfamiliar.

Response: This was explained L 92-96.” There is an increasing interest in using slower-growing and local chickens. Anadolu-T is a registered genotype for broiler production and is within the scope of the selection and breeding program in Türkiye. The pure lines of Anadolu-T and their crosses had lower body weights and impaired feed conversion ratio but higher livability compared to commercial strains [28]. These chickens could be an alternative for niche markets and small local producers”.

Comments Please, add the info about vaccination and any prophylactic program that was used- vitamins, minerals, acidifier (in water). 

What about the coccidiostats - were any of them use in the feed (which one, when?)

Response: The vaccine program was added. Coccidiostats were not used. “This information was added L134-136. “All chicks were vaccinated against Newcastle disease, Gumboro disease, and infectious bronchitis at the hatchery. On d 10 and 18, Newcastle and infectious bronchitis vaccine recalls were performed. Coccidiostats were not used during the experimental period”.

Comments L427- there is no number of the reference, please, add it.,

Response: It was added. L460  ” [48]”

Comments Authors noticed that the response of commercial chickens to the diets was different from that of local chickens. What may be the reason for the effect on blood biochemical parameters? two breeds present two types - slow-growing and long-growing, their metabolism and physiology are slightly different - could this have influenced the results?

Response: Thank you for this comment. The response was added L 470-475 “However, this effect was not observed in commercial broilers. The present study showed that blood parameters of strains differed in their response to diets; i.e. the diets did not affect the blood glucose, ALT, AST, triglyceride, cholesterol, and uric acid levels of commercial chickens. This result might be due to the feeding period which was longer in local chickens than those commercial and/or their different growth rate and genetic background.”.

Comments What, in your opinion (and based on the previous study), is the reason of the observed effect- diets on a weight at day 10? Why, in your opinion there were no effect in older birds? You mentioned about gut health- could you described it in more details?

Response: It was explained L 435-440. ” It is possible to attribute this result to two reasons. 1) The occurrence of initial colonization of the intestinal microbiota has an impact on growth performance and daily gain [51]. BSF larvae meal may have affected the weight gain in this period by contributing to the formation of intestinal microbiota in chicks. 2) The increase in body weight was accompanied by increased feed consumption and appetite of the chicks during the same period. Naturally, chickens like to consume larvae [52].  Heuel et al. [53] reported that the diets containing BSF larvae meal were slightly superior in palatability to a diet including soybean meal, which may increase feed consumption”.

Reviewer 2 Report

Comments and Suggestions for Authors

 In my opinion,  this manuscript is important for broilers industry. However,  here I added some points may add value to this manuscript

1. Title: It is better to add feedstuffs used 

2. Ln11, this sentence not matched with the main of this manuscript

3.Ln 54.  Technical errors in writing

4. Materials and methods 

Please add the chemical composition of sugar and starch in table 2.

5. Results are clearly presented but there are many issues in findings; the values are out of accuracy range especially in commercial breed (cobb). Forexample, FCR and breast and leg meat. FCR ranged nowadays 1.35 - 1.42, please explain the reasons as the FCR in control was 1.7

Comments on the Quality of English Language

Technical errors such as in table 3 ( slaughter) and Ln 54, please check the whole manuscript 

Author Response

We appreciate the time and effort that you have dedicated to providing valuable feedback and comments to improve the quality of the manuscript. I have carefully revised the manuscript and incorporated comments within the manuscript. Changes are highlighted in yellow.

We hope that the revision is satisfactory.

Kind regards

Servet Yalcin

My response to the reviewers' comments

Comments and Suggestions for Authors

 In my opinion,  this manuscript is important for broilers industry. However,  here I added some points may add value to this manuscript

Response: Thank you for offering us to submit the revised manuscript for the opportunity to improve our manuscript.

Comment 1. Title: It is better to add feedstuffs used 

Response: The title was changed.” Effects of partial replacement of soybean with local alternative sources on growth, blood parameters, welfare and economic indicators of local and commercial broilers”

Comment 2. Ln11, this sentence not matched with the main of this manuscript

Response: It was rewritten. L11-15 “While the demand for poultry meat is increasing, arable land for crop production is limited. Therefore, the use of locally produced alternative sources in chicken diets becomes a necessity. The substitution of soybean with 1. local by-products such as sunflower meal, brewers’ dried grain, wheat middlings, or 2. a combination of local by-products with black soldier larvae meal was evaluated in broilers.”

Comment 3.Ln 54.  Technical errors in writing

Response: It was corrected and a dot was included.

Comment 4. Materials and methods 

Please add the chemical composition of sugar and starch in table 2.

Response The table was revised and sugar and starch were included.

Comment 5. Results are clearly presented but there are many issues in findings; the values are out of accuracy range especially in commercial breed (cobb). Forexample, FCR and breast and leg meat. FCR ranged nowadays 1.35 - 1.42, please explain the reasons as the FCR in control was 1.7

 Response The explanation was given L 412-419.” In the present study, the body weights of day-old chicks were similar among the groups indicating that the chicks were distributed randomly. The average slaughter weight and feed conversion ratio were 2247 and 2.33 for local and 2196 g 1.77 for commercial chickens at 55 and 40 d of age, respectively. The results obtained for Cobb chickens were lower than the body weight and feed values specified for Cobb at 40 days [43]. This was because they were fed diets containing less protein than the dietary protein values recommended for Cobb500 (21-22, 19-20, 18-19, 17-18 % protein for starter, grower 1, grower 2, and finisher diets) [44].”

 Comment on the Quality of English Language

Comment Technical errors such as in table 3 ( slaughter) and Ln 54, please check the whole manuscript 

Response: Table 3. It was changed to “Slaughter age”, Ln 54: a dot was included.

Round 2

Reviewer 2 Report

Comments and Suggestions for Authors

Thanks for your revised manuscript

Comments on the Quality of English Language

Fine